# Methane from microbial hydrogenolysis of sediment organic matter before the Great Oxidation Event

Xinyu Xia ⓘ [1✉] & Yongli Gao ⓘ [1]

Methane, along with other short-chain alkanes from some Archean metasedimentary rocks, has unique isotopic signatures that possibly reflect the generation of atmospheric greenhouse gas on early Earth. We find that alkane gases from the Kidd Creek mines in the Canadian Shield are microbial products in a Neoarchean ecosystem. The widely varied hydrogen and relatively uniform carbon isotopic compositions in the alkanes infer that the alkanes result from the biodegradation of sediment organic matter with serpentinization-derived hydrogen gas. This proposed process is supported by published geochemical data on the Kidd Creek gas, including the distribution of alkane abundances, stable isotope variations in alkanes, and $CH_2D_2$ signatures in methane. The recognition of Archean microbial methane in this work reveals a biochemical process of greenhouse gas generation before the Great Oxidation Event and improves the understanding of the carbon and hydrogen geochemical cycles.

[1] Department of Geological Sciences, University of Texas at San Antonio, San Antonio, TX, USA. ✉email: xinyu.xia@utsa.edu

**M**icrobial methane was an important source of greenhouse gases that kept the early Earth warm before the Great Oxidation Event (GOE, 2.4 Ga) under the faint Sun[1]. The generation of microbial methane, as a significant step of the global carbon cycle, is reflected in the stable isotopic compositions of organic matter. However, finding and interpreting records that span billions of years are challenging[2,3]. Over the past two decades, methane and other short-chain alkane gases discovered in ancient shields have shown broad variations in hydrogen isotopic compositions (δD), but relative uniformity in carbon isotopic compositions (δ[13]C)[4,5]. The first reported example is from the Kidd Creek mines in the Neoarchaean Abitibi Greenstone Belt of the Canadian Shield near Timmins, Ontario, Canada[4]. In this area, multiple cycles of sedimentation, volcanic activities, magma intrusions, and serpentinization of mafic rocks occurred ~2.7 Ga[6]. A Neoarchaean ecosystem has been indicated by biomarker evidence[7]. The Kidd Creek mines produce metal sulfides associated with underground felsic volcanic rocks[8]; alkane gases (from methane to pentanes) have been discovered at depths of 2000–3000 m in exploration boreholes[4]. Reported gas contents and isotopic compositions show, in addition to the above δD and δ[13]C patterns, an exponential distribution of alkane abundance and deuterium–deuterium clumped isotopic composition of methane ($\Delta^{12}CH_2D_2$) between −10 and 5‰ (for samples collected <400 days since the borehole was drilled)[4,5,9–11].

Upon its discovery, the Kidd Creek gas was interpreted as abiotic through Fischer–Tropsch (FT) synthesis because the δD and δ[13]C patterns are different from those found in alkanes generated from the decomposition of organic matter[4]. However, these patterns have neither been found in hydrothermal gas, commonly believed as abiotic[12], nor in artificial FT synthetic gas[13]. Therefore, the abiotic interpretation is empirical and lacks a chemical rationale, especially when the feasibility of this synthesis under geological conditions has been questioned in recent years[14–16]. Moreover, the abiotic interpretation overlooked the possibility of forming the unique isotopic distributions in a distinct oxygen-depleting Archaean ecosystem.

In this work, we present the evidence that the Kidd Creek gas is a microbial product in a Neoarchaean ecosystem. From the deuterium distribution in short alkanes, we discover that the hydrogen atoms in each short alkane molecule come from two sources: two capping H atoms are donated by serpentinization-derived $H_2$, while the rest are inherited from long alkyl chains. The distributions of alkane abundances, carbon, hydrogen, and clumped isotopic compositions suggest that the gas was generated through hydrogen biodegradation of sediment organic matter. The Kidd Creek gas is a record of microbial greenhouse gas generation before the GOE.

## Results and discussion

**Identification of alkane precursors.** Figure 1a shows the comparisons of δ[13]C and δD of alkanes between the Kidd Creek gas, typical thermogenic gases[17,18], and gases from a hydrothermal field[12]. The unique contrastive alkane δ[13]C and δD variations in the Kidd Creek gas indicate that the hydrogen isotopic distribution is unlikely due to a pure kinetic isotope effect (KIE) during its formation. Instead, the wide δD variation implies that the hydrogen atoms likely have multiple sources with distinct δD values. The variation suggests a reaction involving the conversion of organic matter to short-chain alkanes in the presence of $H_2$, a hydrogen degradation (or hydrogenolysis) reaction. The exponential distribution of alkane gas abundance with respect to carbon number (Fig. 1b), consistent with a random scission of long alkyl chains[19], also supports the hydrogenolysis explanation.

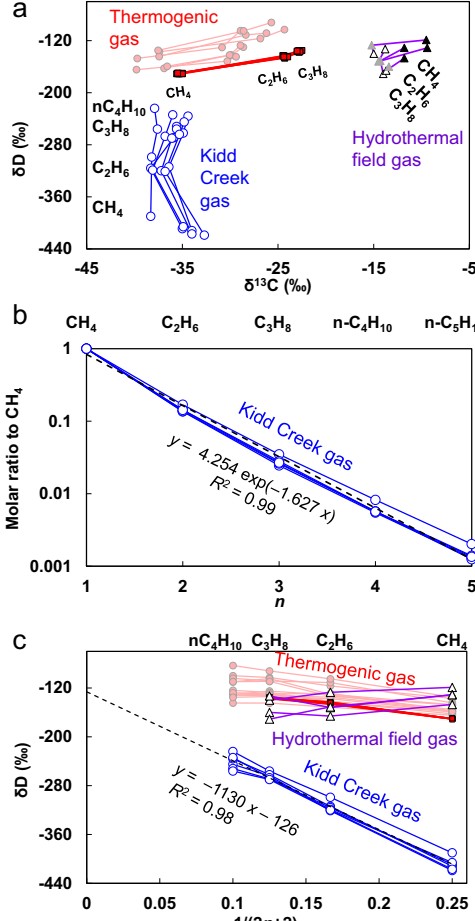

**Fig. 1 Distinct geochemical signatures of the Kidd Creek gas. a** δ[13]C versus δD. **b** Distribution of alkane abundances (relative to $CH_4$). **c** δD against the reciprocal of hydrogen number of alkanes. Blue lines—Kidd Creek gas (depth > 2500 m)[4,5]; violet—hydrothermal field gas data from the Lost City of Mid-Atlantic Ridge[12], black, grey, and white triangles stand for methane, ethane, and propane, respectively; pink—thermogenic gas data from NW Australia Shelf[17]; red—thermogenic gas data from NW Sichuan Basin, China[18]; black dashed lines—linear trend lines of the Kidd Creek gas.

We found a linear correlation between the δD values and the reciprocals of the hydrogen numbers (Fig. 1c), which particularly supports the mechanism of short-chain alkane generation from hydrogenolysis of long alkyl chains, as explained in the following paragraphs.

During hydrogenolysis, long alkyl chains (straight or branched) are consumed segment by segment consecutively. Each generated short-chain alkane molecule may contain 1, 2, or 3 capping hydrogen atoms from $H_2$ depending on the positions of broken C–C bonds (Fig. 2a). Most alkane molecules produced from hydrogenolysis have two capping hydrogen atoms from $H_2$ because secondary carbon atoms are significantly more abundant than primary carbon atoms in a long alkyl chain, especially when one or both ends of the alkyl chain are bonded to a less degradable carbon ring or kerogen. For the degradation of branched C–C chains (in isoprenoid structures), the amount of methane generated from branched primary carbons is equivalent to the amount generated from tertiary carbons; the average number of H atoms from $H_2$ is still 2. As a result, the hydrogen isotopic composition of the generated alkane $C_nH_{2n+2}$ is

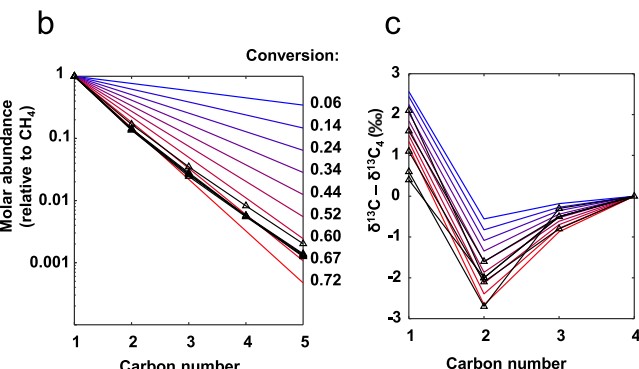

**Fig. 2 Numerical simulation on the distributions of abundance and δ¹³C of short alkanes from hydrogen biodegradation of long alkyl chains. a** Reaction scheme showing that the number of carbon and hydrogen atoms contributed by the alkyl chain and H$_2$ to the gaseous alkanes depends on C–C cleavage positions. Brown, blue, and red represent cases of 1, 2, and 3 capping hydrogen atoms donated by H$_2$, respectively. **b** Numerical simulation on the abundances of gaseous alkanes relative to CH$_4$. **c** Numerical simulation on δ¹³C of gaseous alkanes relative to δ¹³C$_{nC4H10}$. Legend of **b** and **c**: black lines—Kidd Creek gas (samples with both δ¹³C and gas content data)[4,5]; blue, red, and gradient colours in between—modelled results (Eqs. 4–7) with conversion (fraction of CH$_4$-contained ¹²C atoms in all ¹²C atoms) labelled in panel **b**. KIE parameters (identical for hydrogenolyses on kerogen side chains and on alkanes): primary ¹³C KIE of ¹³k/¹²k = 1.0015 (inverse KIE); secondary ¹³C KIE of ¹³k/¹²k = 0.9978 (normal KIE).

expressed as

$$\delta D_{C_nH_{2n+2}} = \frac{2n}{2n+2}\delta D_A + \frac{2}{2n+2}\delta D_B = \delta D_A + \frac{2(\delta D_B - \delta D_A)}{2n+2} \quad (1)$$

In Eq. (1), subscripts A and B refer to hydrogen atoms from the alkyl chain and hydrogen substrate, respectively. Equation (1) shows a linear relationship between $\delta D_{C_nH_{2n+2}}$ and the reciprocals of hydrogen numbers $1/(2n+2)$. Similar analysis has successfully explained the linear variation between δ¹³C and $1/n$ in thermogenic alkanes[20].

A linear covariation with a high coefficient of determination ($R^2 = 0.98$) indeed exists in the Kidd Creek gas (Fig. 1c), yielding a trend line with a $y$-intercept of $\delta D_A = -126$‰ and a slope of $2(\delta D_A - \delta D_B) = -1130$‰ (hence $\delta D_B = -691$‰). The $\delta D_A$ and $\delta D_B$ values are close to the δD values of marine organic matter ($-80$ to $-170$‰)[21] and serpentinization-derived H$_2$ in the Kidd

Creek gas ($\sim -730$‰)[22], respectively. Therefore, the reaction between the long alkyl chains in organic matter and serpentinization-derived H$_2$ explains the δD isotopic pattern of the Neoarchaean alkane gas. The slight difference of $\delta D_B > \delta D_{H2}$ indicates a weak and inverse deuterium KIE (DKIE) during some reaction steps (an inverse KIE means that the substitution of a light isotope by a heavy isotope accelerates the chemical reaction rate). The linear trends with high coefficients of determination in the Kidd Creek gas (Fig. 1b, c) indicate that the interference from other reactions is minimal. These reactions, including the thermal cracking of kerogen, oxidation of methane, and hydrogenolysis of non-alkyl organic precursors (e.g. carbohydrates, which are abundant in living organic matter), yield neither an exponential distribution of short-chain alkane abundance nor the above hydrogen isotope distribution.

**Microbial hydrogenolysis suggested by ¹³C and ¹²CH$_2$D$_2$.** In contrast to the hydrogen atoms, which have two sources, all the carbon atoms in the Kidd Creek hydrocarbon gases originate from organic precursors. As a result, kinetic isotopic fractionation (instead of multiple sources) is the main reason for the δ¹³C variation between short-chain alkanes[20]. This variation is narrow in the Kidd Creek gas (Fig. 1a); therefore, the C–C cleavage has a weak ¹³C KIE, implying that the C–C cleavage is not the rate-determining step[23] and is likely catalytic[24]. The coexistence of C$_{2+}$ alkane gases (>10 vol%) with H$_2$ also suggests a catalytic process. The reason is that a C–C bond (bond energy 346 kJ/mol) is more vulnerable to crack than an H–H bond (bond energy 432 kJ/mol). If there were no catalytic processes to lower the activation barrier of H–H splitting, C$_{2+}$ gases would be more depleted. Because abiotic catalysis requires dispersed elemental transitional metal catalysts without thermal-sintering and sulfur-poisoning[25], a demanding condition hardly satisfied in geological bodies, the catalytic process revealed here is most likely a microbial process.

The order of $\delta^{13}C_{CH_4} > \delta^{13}C_{nC_4H_{10}} > \delta^{13}C_{C_3H_8} > \delta^{13}C_{C_2H_6}$ in most Kidd Creek gas samples (Fig. 1a) suggests a special combination of ¹³C KIEs. A CH$_4$ molecule has only one carbon atom, which experienced a primary ¹³C KIE during C–C cleavage. Other alkane molecules have primary carbon atoms and carbon atoms adjacent to the primary ones. The primary carbon atoms experienced a primary ¹³C KIE during the cleavage; their adjacent ones experienced a secondary ¹³C KIE. The combination of a normal secondary ¹³C KIE and an inverse primary one may form the above δ¹³C order, as demonstrated by a numerical simulation on the consecutive random scission of long alkyl chains (Fig. 2; Model A in 'Methods' section). The simulation yields a perfect exponential distribution of short-chain alkane abundance with carbon number, in addition to the observed δ¹³C pattern (Fig. 2b, c). While an inverse primary KIE is most common for deuterium[26], it is also possible for heavier elements such as ¹³C and ¹⁵N in biochemical reactions[27,28].

Because δD variation in alkanes is mostly determined by the contributions of the two hydrogen sources with distinct δD values, the DKIE during this microbial reaction is hardly evaluable based on δD values. However, the DKIE can be evaluated from the deuterium–deuterium clumped isotopic composition Δ¹²CH$_2$D$_2$. Without a DKIE, a pure stochastic clumped isotopic distribution in the hydrogenolysis reaction would yield a drastically negative Δ¹²CH$_2$D$_2$ value ($-76$‰, for calculation see Model B in the 'Methods' section, after Eq. 14), in contrast to the reported values, which are not far away from zero[10]. One possible explanation for this difference is kinetic relaxation, where the hydrogen exchange between methane isotopologues can bring about an equilibrated Δ¹²CH$_2$D$_2$ value

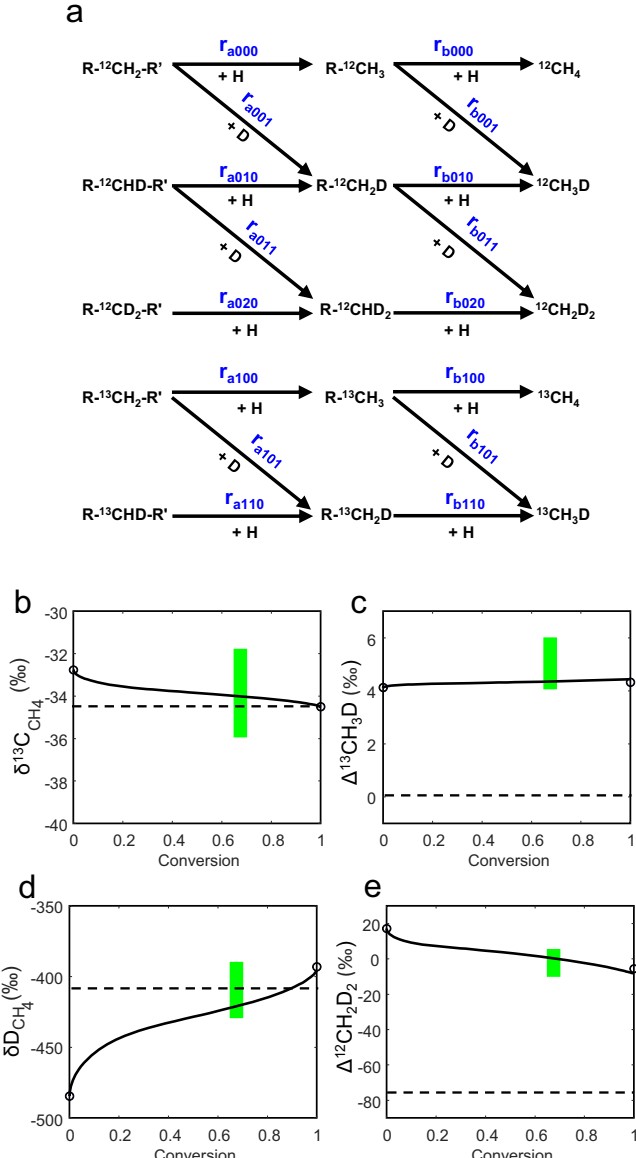

**Fig. 3 Numerical simulation on the isotopic compositions of CH₄ from hydrogen biodegradation of long alkyl chains. a** Reaction scheme showing isotope distribution during the hydrogenolysis of long C–C chains. **b–e** variations of $\delta^{13}C$, $\Delta^{13}CH_3D$, $\delta D$, and $\Delta^{12}CH_2D_2$ with conversion (fraction of CH₄-contained $^{12}C$ atoms in all $^{12}C$ atoms). Legend in **b–e**: dashed lines—kinetic isotope effects (KIEs) are absent and no enrichment/depletion of clumped isotopes in the organic precursor; solid lines—inverse deuterium KIE (DKIE) of the hydrogen donor (primary DKIE) present; circles—analytical solutions (Eqs. 12 and 13) at the start and end of conversion; green bars—range of reported data (conversion is set between 0.65 and 0.70 as constrained in Fig. 2). Parameters are listed in Table 1.

**Table 1 Parameters of numerical simulations on the isotopic fractionation of CH₄ during hydrogen biodegradation (Fig. 3).**

| | No KIE (pure stochastic) | KIE present |
|---|---|---|
| **Reaction rate constant** | | |
| $k_{b000}/k_{a000}$ | Insensitive | 20 |
| **Bulk KIE** | | |
| $\alpha_{ka001}$, $\alpha_{kb001}$ | 1 | 1.10 (inverse 1° DKIE: $k_H/k_D = 0.91$) |
| $\alpha_{ka010}$, $\alpha_{kb010}$ | 1 | 0.9 (normal 2° DKIE: $k_H/k_D = 1.18$) |
| $\alpha_{ka100}$, $\alpha_{kb100}$ | 1 | 1.0009 (inverse $^{13}C$ KIE: $^{12}k/^{13}k = 0.9991$)[a] |
| **Excessive KIE due to D-D clumping** | | |
| $\gamma_{a011}$, $\gamma_{b011}$ | 1 | 1.10 |
| $\alpha_{a011}$, $\alpha_{b011}$ | 1 | 1.0890 |
| $\gamma_{a020}$, $\gamma_{b020}$ | 1 | 1 |
| $\alpha_{a020}$, $\alpha_{b020}$ | 1 | 0.8100 |
| **Excessive KIE due to $^{13}C$–D clumping** | | |
| $\gamma_{a101}$, $\gamma_{b101}$ | 1 | 1 |
| $\alpha_{a101}$, $\alpha_{b101}$ | 1 | 1.1010 |
| $\gamma_{a110}$, $\gamma_{b110}$ | 1 | 1 |
| $\alpha_{a110}$, $\alpha_{b110}$ | 1 | 0.9008 |
| **Initial values of precursors** | | |
| Molar ratio RCH₂R':EH | Insensitive | 1:4 (EH excessive) |
| $\delta^{13}C_A$ (‰) | −34.5 | −34.5 |
| $\delta D_A$ (‰) | −126 | −126 |
| $\delta D_B$ (‰) | −691 | −691 |
| $\Delta R^{13}CHDR'$ (‰) | 0 | 6 |
| $\Delta R^{12}CD_2R'$ (‰) | 0 | 0 |

[a]$\alpha_{ka100}$ and $\alpha_{kb100}$ are lower than the value of 1.0015 obtained in Model A, because Model B does not include a secondary $^{13}C$ KIE.

To demonstrate this explanation, we conducted kinetic numerical simulations on bulk and clumped isotopic fractionations of CH₄ generation (Fig. 3; Model B in the 'Methods' section). Figure 3a shows the scheme of isotope distribution; Fig. 3b–e presents the simulation results with parameters listed in Table 1. The simulation yields an inverse primary DKIE during the generation of Kidd Creek methane ($k_H/k_D = 0.91$), which provides further support that the gas was formed through an enzyme-catalysed microbial process[26].

**Early life before GOE versus deep life after GOE.** Hydrogenotrophy (microbial consumption of H₂) using alkyl chains suggests an abnormally low oxygen fugacity in the ecosystem, where electron acceptors such as $O_2$, $Fe^{3+}$, $SO_4^{2-}$, and even $CO_2$, which are more efficient than alkanes to oxidise H₂, are absent. This anoxic ecosystem could have been widely present either before the GOE or only present in the deep earth after the GOE when the Earth's surface was too oxidative. By comparing hydrocarbon gases from various serpentinization sites (Fig. 4), we found the former condition more plausible:

(1) All previously reported hydrocarbon gas samples with the above alkyl hydrogenolysis features (a widely varied $\delta D$ accompanied by relatively uniform $\delta^{13}C$ in alkanes, and a deuterium-depleted H₂ source as indicated by the $\delta D$ versus reciprocal hydrogen number correlation line) were obtained from the sites with both pre-GOE serpentinization and pre-GOE organic matter sedimentation (Fig. 4). Sedimentation of the Abitibi belt (2.71–2.73 Ga), where the Kidd Creek mines

(close to 0). However, hydrogen atom exchange between methane molecules requires an elevated temperature and the existence of catalysts (free transitional metal in the lab or free radicals under geological conditions); under these conditions, ethane and propane would start to decompose (C₂₊ alkanes <5 vol%)[29]. The high fraction of C₂₊ alkanes (~15 vol%) in the Kidd Creek gas rules out a significant hydrogen exchange between methane isotopologues. A more concrete explanation for the above contrast is that clumped isotopic distribution during methane generation is normally kinetically governed, and an excessive KIE due to D–D clumping may form the observed $\Delta^{12}CH_2D_2$ value[30].

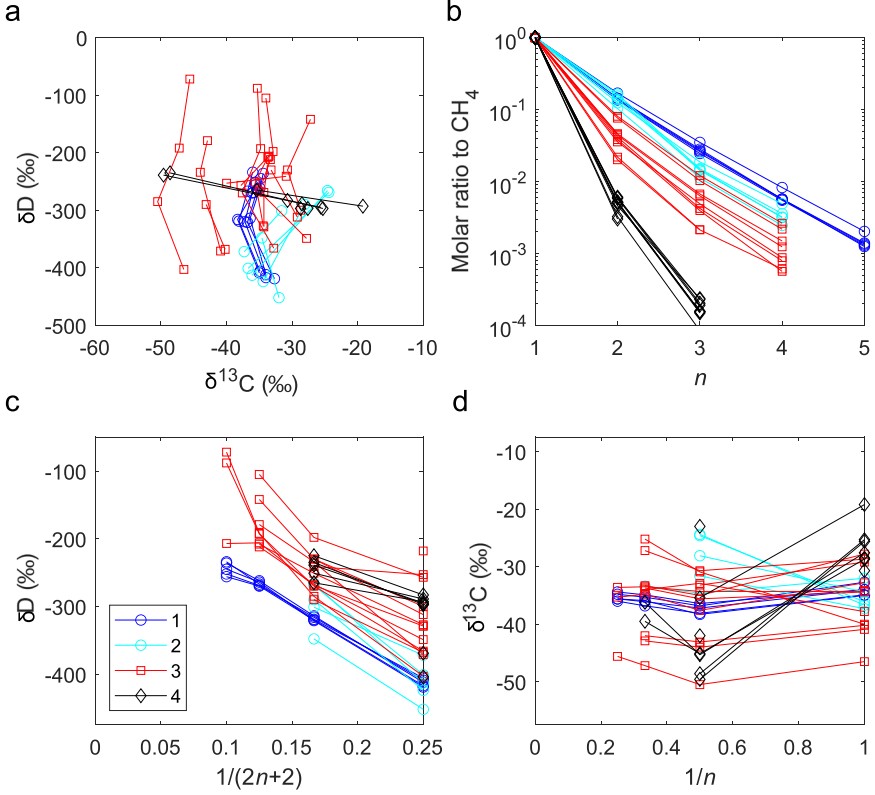

**Fig. 4 Comparison of molecular and isotopic compositions between alkanes from pre-GOE and post-GOE sites. a** $\delta^{13}C$ versus $\delta D$ of alkanes. **b** Exponential (Flory–Schulz) distribution of alkane abundance. **c** $\delta D$ versus reciprocal of hydrogen number. **d** $\delta^{13}C$ versus reciprocal of carbon number. Pre-GOE sites: 1—Kidd Creek mines (2.7 Ga); 2—Copper Cliff mines (2.45 Ga); 3—Kloof, Driefontein, and Mponeng mines (3.0–2.7 Ga). Post-GOE sites: 4—Juuka and Pori mines (1.95 Ga).

are located, was limited between several episodes of volcanic activities and serpentinization in the Neoarchaean. The last volcanic activity (~2.69 Ga[6]) brought about the graphitisation of organic precursors and the termination of biodegradation conditions[31,32]. Other sites include (1) the Kloof, Driefontein, and Mponeng mines in the Witwatersrand Basin of the Kaapvaal Shield (southern Africa), where serpentinization and organic matter sedimentation occurred 3.0–2.7[33], and (2) the Copper Cliff mines in the Sudbury Basin of the Canadian Shield, with sedimentary organic matter in the Paleoproterozoic Huronian Supergroup deposited ~2.45 Ga[34], just before the onset of the GOE[35].

On the contrary, hydrocarbon gas from post-GOE serpentinization sites does not have the above isotopic patterns. These sites range in age from the Proterozoic (the Juuka and Pori mines in the Fennoscandian Shield, with an age of 1.95 Ga[36]) to the present (the Lost City geothermal field, Fig. 1).

(2) When these pre-GOE sites are exposed to modern microbial activity (due to mining), the isotope pattern of hydrogenolysis in the gas eventually becomes altered. The $\delta D$ and $\Delta^{12}CH_2D_2$ values of $CH_4$ in the Kidd Creek gas became more positive years after boreholes were drilled[11]. Microbes have been discovered in the sites of the Witwatersrand Basin[37] and consume propane and $n$-butane, enriching deuterium in the residual of these hydrocarbons for some samples (Fig. 4c).

The above observations show that the occurrence and preservation of isotopic signatures from the hydrogenolysis reaction require a narrow chemical condition and specified microbial species, which are hardly satisfied after the GOE when

strong oxidisers and diversified microbes coexist from shallow to deep earth. These signatures are vulnerable to post-GOE microbial reactions, such as methanogenesis from $CO_2$ and $H_2$ (making the $\delta^{13}C$ more positive), methanotrophy, and the oxidation of higher alkanes. The oxygen-deficient Archaean ecosystem was crucial for the hydrogen biodegradation and the unusual isotopic signatures. Before the GOE, the atmospheric $O_2$ partial pressure was <0.001% or even 0.00001% of the present atmospheric level (PAL); sulfate concentration in the ocean was 2.4% of the current level[38]. The oxygen-deficient atmosphere and ocean prevented both biological and chemical oxidation of organic matter in sediments[39]. Therefore, the Kidd Creek gas is a chemical fossil of the pre-GOE ecosystem preserved in the tectonically stable Canadian Shield.

The microbial hydrogenolysis process explains the $^{13}C$ depletion in solid organic matter in some Archaean metasedimentary rocks. There are two distinct types of $\delta^{13}C$ values in Archaean solid organic matter: one is within the normal range (−26 to −33‰), and the other is more depleted (−35 to −44‰, including the kerogen in the Abitibi belt of −43.8‰)[40]. The latter was attributed to methane cycling[3], where methane was considered a precursor of the kerogen. Our discovery provides a more plausible explanation: kerogen (and its precursor) in some Archean sediments may have undergone microbial hydrogenolysis in the presence of $H_2$. Through this alkyl removal process with a weak and possibly inverse $^{13}C$ KIE, the residual kerogen (and the kerogen formed by the residual precursor) may have been depleted of $^{13}C$ and side alkyl chains. Due to the lack of side alkyl chains, the residual kerogen would not have experienced $^{13}C$ enrichment by thermal dealkylation; it would have kept the $^{13}C$-depleted signature.

The long preservation time of a pre-GOE gas accumulation (2.7 Gyr for the Kidd Creek gas) raises the question of whether the isotopic compositions of the residual gas were affected by diffusive fractionation. This impact is insignificant according to the two comparisons:

(1) Comparison between the Kidd Creek rocks and typical Palaeozoic natural gas reservoirs on their diffusion time scales. The characteristic time scale ($\tau$) for diffusion is inversely proportional to diffusivity[41]. Diffusive fractionation starts to have a significant impact on the isotopic compositions of residual gas when the preservation time is close to the characteristic time[42]. The sealing rocks of the Kidd Creek gas are crystalline rocks with gas diffusivity ($<10^{-15}$ m$^2$/s) three orders of magnitude lower than tight shale ($10^{-12}$ m$^2$/s)[42–44], meaning that $\tau$ of the Kidd Creek gas-bearing rocks is 1000 times longer than that of the typical Palaeozoic gas reservoir in shale rocks, compensating their 10-fold difference in the preservation time. There is no significant impact from diffusive fractionation through shales on the isotopic compositions of the Palaeozoic natural gas accumulations[44]; therefore, the residual Kidd Creek gas unlikely experienced remarkable diffusive fractionation.

(2) Comparison between H$_2$ and alkanes on isotopic fractionation. H$_2$ has a smaller molecular mass and is more sensitive to diffusive fractionation than alkanes. The δD values of H$_2$ in the Kidd Creek gas are within the range of modern serpentinization-derived H$_2$[22], indicating a minimal influence of diffusion on the isotopic compositions of the alkanes.

Hydrogenolysis of organic matter could act as a pathway for Archaean microbial life to obtain chemical energy stored in H$_2$. Considering bond energies of 346, 411, and 432 kJ/mol for C–C, C–H, and H–H bonds, hydrogenolysis of long-chain alkanes is an exothermic reaction with a net heat release of 44 kJ/(mol H$_2$). In contrast to the modern hydrogenotrophic methanogenesis of oxygen-bearing compounds[45], microbial hydrogenolysis of long alkyl chains is a different geochemical pathway to convert H$_2$ and acted as a significant source of greenhouse gas before the GOE, when the young Earth experienced widespread serpentinization of mafic volcanic rocks in the high heat-flow lithosphere. Conversion of organic matter and the serpentinization-derived H$_2$, along with the hydrogen isotopic distribution in the ecosystem, are summarised in Fig. 5.

## Methods

**Model A: abundances and δ$^{13}$C of short alkanes.** Considering the cleavage at position $m$ (between the no. $m$ and no. $m+1$ carbon atoms) of an $n$-alkyl chain with $n$ carbon atoms ($1 \le m < n$), the hydrogenolysis reaction equation is

$$\text{R-C}_n\text{H}_{2n+1} \xrightarrow{2[\text{H}]} \text{R-C}_m\text{H}_{2m+1} + \text{H-C}_{n-m}\text{H}_{2(n-m)+1} \qquad (2)$$

Similarly, the reaction equation for cleavage on an $n$-alkane molecule is

$$\text{H-C}_n\text{H}_{2n+1} \xrightarrow{2[\text{H}]} \text{H-C}_m\text{H}_{2m+1} + \text{H-C}_{n-m}\text{H}_{2(n-m)+1} \qquad (3)$$

Branched alkyl chains are omitted for simplicity. The cleavage reactions are consecutive; the products R-C$_m$H$_{2m+1}$, H-C$_m$H$_{2m+1}$, and H-C$_{n-m}$H$_{2(n-m)+1}$ (written below as RC$_m$, HC$_m$, and HC$_{n-m}$) are still subject to hydrogenolysis until all C–C bonds break down with CH$_4$ as the ultimate product. The cleavage rate ($r$) of a kerogen side chain (RC$_m$) or an alkane molecule (HC$_m$) is proportional to the C–C chain length and concentration $c$:

$$r_{\text{RC}_m} = -mkc_{\text{RC}_m} \qquad (4)$$

$$r_{\text{HC}_m} = -(m-1)kc_{\text{HC}_m} \qquad (5)$$

Here, $k$ is the reaction constant to break any C–C bond. For simplicity, the difference in $k$ between a middle and an end bond in the C–C chains is not considered. The net reaction rate of a kerogen side chain with a length of $m$ carbon atoms is

$$\frac{dc_{\text{RC}_m}}{dt} = -mkc_{\text{RC}_m} + \sum_{n=m+1}^{N}(kc_{\text{RC}_n}) \qquad (6)$$

Here, $t$ is time and $N$ is the maximum chain length of the reaction system. The first term of the right-hand side accounts for the cleavage of the kerogen side chain, and the second term accounts for the generation of residual shorter side chains from the cleavage of the longer side chain. The net reaction rate of a normal alkane with $m$ carbon atoms is

$$\frac{dc_{\text{HC}_m}}{dt} = -(m-1)kc_{\text{HC}_m} + \sum_{n=m}^{N}(kc_{\text{RC}_n}) + 2\sum_{n=m+1}^{N}(kc_{\text{HC}_n}) \qquad (7)$$

The first term of the right-hand side accounts for the cleavage of this alkane, the second term accounts for the generation of the alkane from the kerogen side chain, and the third term accounts for the generation from the cracking of alkanes longer than this alkane. The factor in the last term is necessary because HC$_m$ is generated from cuttings at the $m$ and the $n-m$ positions of HC$_n$.

Equations (6) and (7) show that shorter chains become more enriched as the cracking goes on. On the one hand, longer chains are more prone to cracking because they have more C–C bonds; on the other hand, shorter chains are the products of longer chains.

Both primary and secondary $^{13}$C KIEs on the rate constant $k$ are considered. Suppose that there is a $^{13}$C substitution at position $j$ ($1 \le j \le n$); if $j = m-1$ or $j = m$, then there is a primary $^{13}$C KIE, or, if $j = m-2$ or $j = m+1$, there is a secondary $^{13}$C KIE on the cleavage at the $m$ position.

A normal distribution is used for the molar distribution of the lengths of initial kerogen side chains, with a minimum chain length of $n_{\min}$, a maximum chain length of $n_{\max}$, a mean length of $(n_{\min} + n_{\max})/2$, and a standard deviation of $\sigma$. When the mean length is large enough ($>15$), the isotopic compositions of gas products are insensitive to the initial kerogen side chain length distribution. For initial values, a δ$^{13}$C value of $-35$‰ is applied. The initial chain length is in a normal distribution with a peak of C$_{17}$ and a standard deviation of $\sigma = 2$ carbon atoms. The initial alkane concentrations are assumed to be 0.

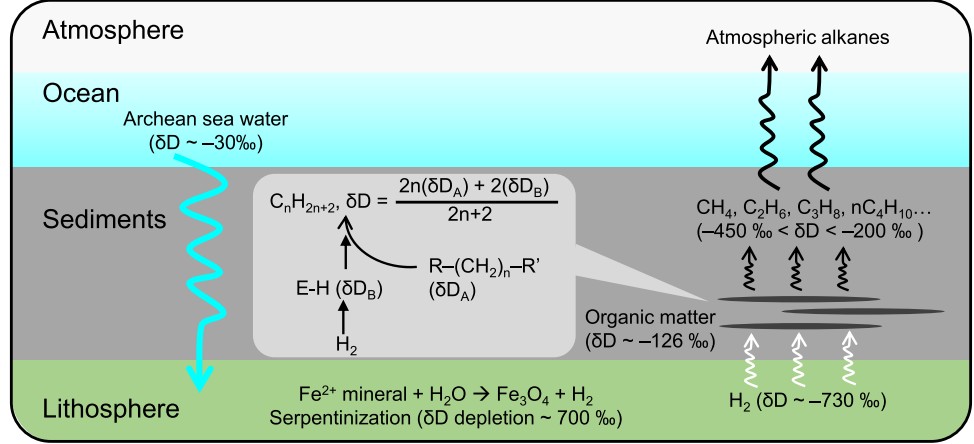

**Fig. 5 Conceptual model.** The model is showing hydrogen isotope fractionation during alkane generation from the biodegradation of organic matter with serpentinization-derived hydrogen in the Neoarchaean ecosystem.

For simplicity, we assume that since there is no isotopic fractionation within or between the alkyl chains at the beginning of hydrogenolysis, the probability of $^{13}C$ substitution at any position of any side chain is identical and determined by the initial carbon isotopic composition $\delta^{13}C$. Multiple $^{13}C$ substitutions on a C–C chain are omitted because consideration of multiple substitutions would drastically increase the modelling complexity. This approximation is valid when the C–C chain is not too long. For example, the ratio between the probabilities of double and single $^{13}C$ substitution in a $C_{20}$ chain is $\left[\binom{20}{2}\left(\frac{^{13}C}{^{12}C}\right)^2\right] / \left[\binom{20}{1}\left(\frac{^{13}C}{^{12}C}\right)\right] \approx 10\%$ for $^{13}C/$ $^{12}C \sim 0.01$. Such a chain is long enough that the $\delta^{13}C$ of gas products is insensitive to C–C chain length. Numerical simulation was conducted with Mathworks MATLAB 2020a.

**Model B: bulk and clumped isotopic fractionations of $CH_4$.** Conversion of methylene in a long C–C chain to methane is generalised into two steps:

$$\text{R-CH}_2\text{-R'} \xrightarrow[+H]{r_a} \text{R-CH}_3 \xrightarrow[+H]{r_b} \text{CH}_4 \tag{8}$$

The first step (step *a*) is the conversion of the methylene group R-CH₂-R' to a

$$
\begin{cases}
\delta^{13}C_{CH_4}\big|_{f=1} = \alpha_{ka100}\alpha_{kb100}\delta^{13}C + (\alpha_{ka100}\alpha_{kb100} - 1) \\[2mm]
\delta D_{CH_4}\big|_{f=1} = \left(\frac{\alpha_{ka010}\alpha_{kb010}}{2}\delta D_A + \frac{\alpha_{ka001}\alpha_{kb010} + \alpha_{kb001}}{4}\delta D_B\right) + \left(\frac{\alpha_{ka010}\alpha_{kb010}}{2} + \frac{\alpha_{ka001}\alpha_{kb010} + \alpha_{kb001}}{4} - 1\right) \\[2mm]
\Delta^{13}CH_3D\big|_{f=1} = \frac{\alpha_A^B[\alpha_{ka001}\alpha_{kb010}(\gamma_{a101}\gamma_{b110} - 1) + \alpha_{kb001}(\gamma_{b101} - 1)] + 2\alpha_{ka010}\alpha_{kb010}[(1+\Delta R^{13}CHDR')\gamma_{a110}\gamma_{b110} - 1]}{\alpha_A^B(\alpha_{ka001}\alpha_{kb010} + \alpha_{kb001}) + 2\alpha_{ka010}\alpha_{kb010}} \\[2mm]
\Delta^{12}CH_2D_2\big|_{f=1} = \frac{8[(\alpha_A^B)^2\alpha_{ka001}\alpha_{kb001}\alpha_{kb010}\gamma_{b011} + 2\alpha_A^B\alpha_{ka010}\alpha_{kb010}(\alpha_{ka001}\alpha_{kb010}\gamma_{a011}\gamma_{b020} + \alpha_{kb001}\gamma_{b011}) + \alpha_{ka010}^2\alpha_{kb010}^2\gamma_{a020}\gamma_{b020}]}{3[\alpha_A^B(\alpha_{ka001}\alpha_{kb010} + \alpha_{kb001}) + 2\alpha_{ka010}\alpha_{kb010}]^2} - 1
\end{cases}
\tag{12}
$$

methyl group (RCH₃) by accepting a capping hydrogen atom from the hydrogen donor (activated H₂); the second step (step *b*) is the conversion of the methyl group to methane by accepting another capping hydrogen atom. This scheme is highly generalised, and each step may involve multiple elementary biochemical reaction steps, such as the binding of H₂ and long alkyl chains to the enzyme, activation of H–H and C–C bonds, and release of the short alkane products from the enzyme. It is beyond the scope of this work to discuss the detailed biochemical reaction steps. But the cleavage and formation of chemical bonds in these steps should be constrained by the observed isotopic patterns.

Due to the computational complexity, we did not use the random scission model (Model A) in the simulation involving clumped isotopic fractionation, as explained in the following. A conventional kinetic model of the decomposition of organic matter without considering the constraints of C–C chain lengths is a zero-dimensional problem. Modelling the random cutting of long C–C chains without considering isotopes is a one-dimensional problem, and modelling bulk carbon isotopic fractionation during random cutting (Model A) is a two-dimensional problem. If $^{13}C$–$^{13}C$ coupling is included in random cutting, the modelling is a three-dimensional problem; a complex Monte Carlo method has been applied to deal with this problem[19]. If the $^{13}C$–D or D–D coupling is included in Model B, as we wish, it is a problem above the fourth dimension. The complexity of programming and the difficulty of computation make the model unattainable; even if it is achievable, solving this problem is far beyond the scope of this work.

Reaction equation Eq. 8 is expanded to the scheme in Fig. 3a to quantify the five most abundant isotopologues in methane (three or more substitutions such as $^{13}CH_2D_2$ or $^{12}CHD_3$ are ignored due to their low abundances). For the subscripts in Fig. 3a (*m*, *i*, and *j* in $r_{amij}$ or $r_{bmij}$), the first digit (*m* = 0 or 1) is the number of $^{13}C$ atoms involved in the reaction, the second digit (*i* = 0, 1, or 2) is the number of deuterium atoms connected in the methylene or methyl group, and the third digit (*j* = 0 or 1) is the number of deuterium atoms in the hydrogen donor.

Clumped isotopic compositions of methylene and methane are defined as the following:

$$
\begin{cases}
\Delta R^{13}CHDR' = \frac{(R^{13}CHDR')(R^{12}CH_2R')}{(R^{13}CH_2R')(R^{12}CHDR')} - 1 \\[2mm]
\Delta R^{12}CD_2R' = 4\frac{(R^{12}CD_2R')(R^{12}CH_2R')}{(R^{12}CHDR')^2} - 1 \\[2mm]
\Delta^{13}CH_3D = \frac{(^{13}CH_3D)(^{12}CH_4)}{(^{12}CH_3D)(^{13}CH_4)} - 1 \\[2mm]
\Delta^{12}CH_2D_2 = \frac{8}{3}\frac{(^{12}CH_2D_2)(^{12}CH_4)}{(^{12}CH_3D)^2} - 1
\end{cases}
\tag{9}
$$

Note that the isotopic compositions here are expressed in decimals; they should be multiplied by 1000 to give per mil values.

The deuterium isotope ratio between the hydrogen donor (denoted with subscript B) and the methylene group (subscript A) is expressed as:

$$\alpha_A^B = \frac{1 + \delta D_B}{1 + \delta D_A} \tag{10}$$

For each reaction step in Fig. 3a, the corresponding rate constants are denoted as $k_{amij}$ for step *a* or $k_{bmij}$ for step *b*. Kinetic fractionation factors $\alpha_{kamij} = k_{amij}/k_{a000}$ and $\alpha_{kbmij} = k_{bmij}/k_{b000}$ define KIEs. Note that a DKIE is often expressed as $k_H/k_D$, which is the reciprocal of the $\alpha_k$ nomenclature here. A DKIE may be primary or secondary; a primary DKIE results in $\alpha_{ka001} \neq 1$ and $\alpha_{kb001} \neq 1$, and a secondary one results in $\alpha_{ka010} \neq 1$ and $\alpha_{kb010} \neq 1$. Kinetic clumped isotope fractionation factors $\gamma_{amij} = \alpha_{kamij}/(\alpha_{ka100}{}^m\alpha_{ka010}{}^i\alpha_{ka001}{}^j)$ and $\gamma_{bmij} = \alpha_{kbmij}/(\alpha_{kb100}{}^m\alpha_{kb010}{}^i\alpha_{kb001}{}^j)$ define the excessive KIE due to isotope clumping in steps *a* and *b*, respectively[30].

Conversion of the reactant R-CH₂-R' is defined as $1 - f$, where $f$ is the residual fraction of R-CH₂-R':

$$f = (\text{R-CH}_2\text{-R'})/(\text{R-CH}_2\text{-R'})_{initial} \tag{11}$$

Considering the isotope abundance of D << H, the analytical solution of isotopic compositions at the beginning of the reaction (*f* = 1) is derived as:

If the abundance of the hydrogen donor is excessive and thus approximately constant, then the analytical solution at the end of the reaction (*f* = 0) is

$$
\begin{cases}
\delta^{13}C_{CH_4}\big|_{f=0} = \delta^{13}C_A \\[2mm]
\delta D_{CH_4}\big|_{f=0} = \left(\frac{1}{2}\delta D_A + \frac{\alpha_{ka001} + \alpha_{kb001}}{4}\delta D_B\right) + \left(\frac{\alpha_{ka001} + \alpha_{kb001}}{4} - \frac{1}{2}\right) \\[2mm]
\Delta^{13}CH_3D\big|_{f=0} = \frac{\alpha_A^B[\alpha_{ka001}(\gamma_{a101} - 1) + \alpha_{kb001}(\gamma_{b101} - 1)] + 2\Delta R^{13}CHDR'}{\alpha_A^B(\alpha_{ka001} + \alpha_{kb001}) + 2} \\[2mm]
\Delta^{12}CH_2D_2\big|_{f=0} = \frac{8[(\alpha_A^B)^2\alpha_{ka001}\alpha_{kb001}\gamma_{b011} + 2\alpha_A^B(\alpha_{ka001}\gamma_{a011} + \alpha_{kb001}\gamma_{b011}) + 1]}{3[\alpha_A^B(\alpha_{ka001} + \alpha_{kb001}) + 2]^2} - 1.
\end{cases}
\tag{13}
$$

If the KIE is absent (all $\alpha_k$ and $\gamma$ factors equal to unity) and the clumped isotopic compositions in the precursor are 0, then clumped isotopic compositions become the following:

$$
\begin{cases}
\Delta^{13}CH_3D = 0 \\[2mm]
\Delta^{12}CH_2D_2 = -\frac{1}{3}\left(\frac{1 - \alpha_A^B}{1 + \alpha_A^B}\right)^2
\end{cases}
\tag{14}
$$

With $\alpha_A^B = 0.354$ from $\delta D_A = -126‰$ and $\delta D_B = -691‰$ given in the text, $\Delta^{12}CH_2D_2 = -76‰$ is obtained. Both $\Delta^{13}CH_3D$ and $\Delta^{12}CH_2D_2$ are more depleted of clumped isotopes than the reported values (4–6‰ for $\Delta^{13}CH_3D$ and −10 to 5‰ for $\Delta^{12}CH_2D_2$)[10,11], indicating that $^{13}C$–D clumping in the methylene precursor should be considered to explain $\Delta^{13}CH_3D$, and DKIE should be considered to explain $\Delta^{12}CH_2D_2$.

Numerical simulations are carried out to find parameters satisfying the following:

(1) The $\delta^{13}C_{CH4}$, $\delta D_{CH4}$, $\Delta^{13}CH_3D$, and $\Delta^{12}CH_2D_2$ values at higher conversions of organic precursors are within the reported ranges.
(2) The $\delta D_A$ and $\delta D_B$ values are close to the derived values from Fig. 1c (−691 and −126‰, respectively) to show that $\delta D_{CH4}$ is mainly determined by $\delta D$ of the precursors rather than by DKIE during the hydrogenolysis.

A value of $\Delta R^{13}CHDR'$ =6‰ in the organic precursor is applied so that the final $\Delta^{13}CH_3D$ is in the range of reported values. This $^{13}C$–D clumping in the precursor is acceptable, considering that $\Delta^{13}CH_3D = 5.6‰$ has been reported for biogenic gas[10,11]. A $\Delta^{12}CH_2D_2$ value close to the observed value but much higher than the stochastic one (Eq. 14) requires $\gamma_{a011} > 1$ or $\gamma_{b011} > 1$, as shown by the $\Delta^{12}CH_2D_2$ expression in Eq. (13). With this prerequisite, either an inverse primary DKIE (1° DKIE, $\alpha_{ka001} > 1$, $\alpha_{kb001} > 1$) or an inverse secondary DKIE (2° DKIE, $\alpha_{ka010} > 1$, $\alpha_{kb010} > 1$) is necessary, and through numerical simulation, we found

that only the inverse 1° DKIE satisfies the above-mentioned $\delta D_A$, $\delta D_B$, and $\Delta^{12}CH_2D_2$ values.

Two scenarios (one is the pure stochastic condition, the other is with an inverse 1° DKIE) are modelled (Fig. 3). The parameters are listed in Table 1. For comparison, analytical solutions at the beginning and end of reactions from Eqs. (12) and (13) are presented. The numerical and analytical solutions are nearly identical at the beginning of conversion. There are small differences between the numerical and analytical solutions at the endpoint because the abundance of the hydrogen donor is not extremely excessive. A weak $^{13}C$ fractionation between the organic precursor and the methane product is obtained with the KIE parameters (Fig. 3b). With such a weak $^{13}C$ KIE, $\Delta^{13}CH_3D$ is nearly constant for reaction extent (Fig. 3c). Note that we applied an inverse $^{13}C$ KIE, as required by the $\delta^{13}C$ distribution of the alkane gases (Method 1, Model A). The $\delta D$ and $\Delta^{12}CH_2D_2$ values are independent of $^{13}C$ KIE. Both the bulk and clumped isotopic compositions of methane within the range of reported values are obtained at the organic precursor conversion of 0.65–0.70 as constrained by Fig. 2.

## Data availability
The datasets generated and/or analysed during the current study are available from the corresponding author on reasonable request.

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

## Acknowledgements
This work was supported in part by the endowment of Amy Shelton and V.H. McNutt Distinguished Professorship in Geology at UTSA.

## Author contributions
X.X. and Y.G. analysed data and wrote the paper.

## Competing interests
The authors declare no competing interests.
