## [Peer Review File · Nature Communications]

REVIEWER COMMENTS

Reviewer #2 (Remarks to the Author):

This is my fourth review of a version of this manuscript. In my last review, I commented on the disagreement between the two models presented in Methods sections 1 and 2. Specifically, the authors have created two independent models for different aspects of the same process: the enzymatically-catalyzed cracking of longer n-alkanes into shorter n-alkanes, and the two-step conversion of methylene groups into methane. I previously pointed out at least two ways in which the models are not self-consistent. I.e., the two models require different values for the same parameters (the reaction progress and the carbon-13 content of methane). I also recommended that the authors provide more documentation on their second model, because this would provide another way to test whether these models are in agreement. For the details of the ways that the two models disagree, please see my previous review.

In response to these comments, the authors have added a paragraph describing why two models are needed, because a single model that predicts all parameters would be computationally expensive and under-constrained. I agree with this assessment. However, it appears that the authors have not addressed the underlying issue: the two models still disagree on the same constraints. So, my previous recommendation still applies: "The present situation is simpler, more straightforward, but results in two models for the same process that appear to be inconsistent. Even so, given that these models and concepts are tangential to the main findings of the paper, this may now be "close-enough." The authors have probably done enough to show that it would be possible to create a single model for microbial hydrogenolysis that explains all aspects of the Kidd Creek data. That such a model does not currently exist could be beside the point...at this stage, perhaps it would be better to publish this and let readers decide for themselves."

Reviewer #4 (Remarks to the Author):

Review of: Methane from microbial hydrogenolysis of sediment organic matter before the Great Oxidation Event

Summary/Key Results:

In this study the authors analyze previously generated carbon and hydrogen isotope data from the Kidd Creek Mine, from a Neoproterozoic Greenstone Belt in Ontario, Canada. The significance of the samples, is that they are some of the oldest occurring hydrocarbons and they have very depleted hydrogen isotope values, making their interpretation difficult. Previous studies have suggested that the complex isotope patterns are a result of Fischer-Tropsch synthesis, but these authors suggest it is a biotic signature. The authors call on hydrogenotrophy using the alkyl chains to help in the processes as the major metabolism imparting the very depleted hydrogen isotope values in this system. If true, this would be of great significance because it may be the oldest record of hydrogenotrophic metabolism preserved in a geological record.

Validity:

There are some large logical leaps which need further justification. While there is no strong evidence that counters the conclusions that the authors draw, there are more simple explanations that the authors have not considered and countered.

The major concern with the study is that the biogenic origin of the $\delta^{13}\text{C}$ pattern is not sufficiently justified. The authors claim that resultant isotope fingerprint can only occur as a result of metabolism. However, the most canonical treatment of hydrocarbons (Chung 1988 and many works that follow) allows for a much simpler explanation of the data. In general, it is predicted for a linear decrease in $\delta^{13}\text{C}$ vs $1/n$ (where n is the carbon number of the alkane in the system). If the methane deviates

from the line of butane, propane, ethane below it is thought to be due to microbial activity. If it deviates above (like it seems as if this data does) it is thought to be from diffusive loss. While this deviation is much larger than in other studies, there could be diffusive loss of methane for ~3 billion years. While this would indicate a very large initial reservoir of methane, it should be calculated and modeled. This is a very traditional and useful representation of hydrocarbon isotope values, and as it would also be a very simple and intuitive answer to the complex data, it needs to be ruled out before more complex interpretation can continue.

There is another line of evidence that the others propose which can be explained more simply. The authors use the $\Delta^{12}\text{CH}_2\text{D}_2$ near zero as evidence of biological fractionation (when most biological fractionation significantly depletes this measurement). Due to the extremely long storage time, many reactions that are kinetically limited can occur (like the re-ordering of deuterium isotopes between methane molecules). There is a recent study by Labidi et al looking at the relative $^{13}\text{CH}_3\text{D}$ vs $^{12}\text{CH}_2\text{D}_2$ of methane at mid ocean ridges, that show that the $^{12}\text{CH}_2\text{D}_2$ can lose its depletion relative to expected equilibrium very quickly. The authors use their simple kinetic model of a $\Delta^{12}\text{CH}_2\text{D}_2 \approx 0$ to indicate their preferred methane metabolism, when other authors have shown significant relaxation of the $\Delta^{12}\text{CH}_2\text{D}_2$ depletion on "modern" timescales at vents. While this not be what is occurring in Kidd Creek, it needs to be addressed as it is the simpler explanation to the isotope value.

The other easy justification of the $\Delta^{12}\text{CH}_2\text{D}_2$ value is the combinatorial effect, which has been documented theoretically and experimentally in recent studies. This allows for a range of clumped values to occur with depleted hydrogen, especially if there are two disparate pools of hydrogen contributing to the methane isotope value. As the dD of the methane is not as low as the dD of the H_2 from serpentinization, this would indicate that the hydrogen addition could in addition to the inheritance from the precursor molecule, could also come from two disparate pools.

Furthermore, while oxygen was likely quite low in the Neoproterozoic, the authors need to indicate the oxygen range that they are speaking about in order for this to add to the discussion of early earth surface conditions. The authors say that it had to be "low" in order for the metabolism they call on to occur, and that this does not happen in more recent serpentinization environments. As this argument hinges on the low $p\text{O}_2$ of the Archean it is necessary to explain the oxygen threshold (either in PAL or fugacity as it is within the crust) that these reactions occur at. Also, serpentinites have a range of fugacity's (and resultant $\text{Fe}^{3+}/\text{Fe}^{\text{total}}$ values) as is dependent on the fluids flushing the system. Therefore, it is more important to declare a value as the range is large.

Finally, strongly reconsider the title away from GoE and focus on interpretation of the deep mine methane in context to the geochemical environment of the mine system and diagenetic processes (see Giunta et al.; Sherwood-Lollar et al. and other studies of bulk and clumped isotope systematics from those samples).

Significance:

If the conclusions of the paper are true, then the paper does have sufficient significance to be published in Nature Communications. Evidence of metabolism that predate the GOE are always very important to understanding the rise of life on Earth. In addition, this metabolism can only occur in very low oxygen environments which also further adds to earth surface constraints at a time when little data exists. Finally, the presence of actual biologically produced products that are Neoproterozoic in age is also significant by itself.

Data and Methodology:

The study does not create any data. All measurements listed in the figures are from previously published work. This is somewhat disappointing, as there are a lot of new measurements in methane isotopes that could perhaps further constrain this complex system. While the authors do bring in the

clumped isotope results that already exist from this well studied local (including both $^{13}\text{CH}_3\text{D}$ and $^{12}\text{CH}_2\text{D}_2$), their indicators could be taken more into account.

Analytical Results:

The authors do generate a novel model with available published data, and provide a new interpretive framework for samples from systems such as the Kidd Creek mine. The model approach could be extrapolated to other available environmental data (see recent studies from Labidi et al.; Giunta et al.; and others). Specifically, diffusive modeling and kinetic relaxation of the depleted $\Delta^{12}\text{CH}_2\text{D}_2$ would be beneficial additions. In addition, the Chung-type plots as requested above would help the reader place into context the new advance made herein.

Suggested improvements:

The paper needs three major additions:

- a) "Chung"-like diagrams made (or fig 1 modified) such that traditional interpretation of the carbon isotope values can be completed easily (if it does not have conclusive results can be included in supplement).
- b) Modeling of diffusive loss of methane and its effect on the deuterium isotope value.
- c) Modeling of kinetic relaxation of $\Delta^{12}\text{CH}_2\text{D}_2$ clumping in methane
- d) Discussion of oxygen fugacity or at least relative to PAL.

Clarity and context:

The text needs to be proofread. There are many errors that decrease clarity of the arguments. Specific examples are included in the line comments, but in general the language could use work.

References:

Sufficient, though there are some more recent papers emerging from both the Eiler lab at CalTech and the Young lab at UCLA that have dealt with methane from a variety of environments.

Expertise:

- A. Piasceki: Stable isotope geochemist that has done work on site-specific and clumped isotopes within small n-alkanes from modeling, methods development, and measurement.
- B. W. Leavitt: Geomicrobiology & microbial geochemistry.

Specific Line comments

- Line 71: why do you have 2 hydrogens per cleavage and not one (theoretically this is the same a thermogenic gas generation over long term, how specifically is this different? unclear besides the incredibly depleted H2 value from serpentinization specifically why biology must be introduced) (discussed above as well)
- Line 92 : sentence structure is ambiguous "because organic precursor is the single carbon source" what does this mean, do you have a single organic carbon precursor (if yes how do you know this, there is usually a diversity of molecules even if from one environment), and the sentence continues to say "kinetic isotope fractionation is the main reason for ... variation within short alkanes", which isn't strictly true, but it does emphasize preexisting structures within the precursor molecules, and overprints an isotope fractionation on top of that
- line 101: does not fully justify the biology of cracking, following the early stolper papers we have some generation of methane at high temperatures, its just how long are the samples within the cracking window, yes you still have higher n-alkanes but this doesn't rule out thermogenic generation?
- line 131 doesn't make sense "where oxidizers that are more efficient than alkanes" are alkanes

oxidizers? or is it what the bugs do?

- line 156: sentence needs modification (became more positive years after boreholes were drilled? need verb)
- following sentence "microbes has" -> microbes have

Reference Cited:

Chung, H.M., Gormly, J.R. and Squires, R.M., 1988. Origin of gaseous hydrocarbons in subsurface environments: theoretical considerations of carbon isotope distribution. *Chemical Geology*, 71(1-3), pp.97-104.

Reply to Reviewer 2

We thank Reviewer 2 again for the long effort in thoroughly reviewing and evaluating our work. Reviewer 2's insightful comments have been valuable to improve the quality of this work. There are two issues remained in the third and fourth reviews:

1. Provide more documentation on the second model.

We submitted spreadsheets containing the time-lapse abundances of all the modeled species from Model A (simulating carbon isotopic fractionation of all alkanes) and Model B (simulating clumped isotopic fractionation of methane).

2. Explaining that the two models still disagree on the same constraints.

The comments in the third review are listed and replied point-to-point in the following.

Comment: *The first disagreement concerns reaction progress. Model A accurately recovers the molar abundance and isotopic compositions of Kidd Creek n-alkanes at a kerogen conversion fraction of 0.8, and no higher. At higher conversion fractions, too little C₂–C₅ n-alkanes would remain. However, model B only recovers observed values at a conversion fraction exceeding 0.8. Shouldn't these conversion fractions agree? If so, there exists an awfully small window where this is the case.*

Reply: This insightful comment helps us improve the calculation on conversion of the organic precursor. In the previous version we did not apply the same definition of conversion in both models and did not use the more rigorous conversion range and ¹³C KIE from Model A to constrain Model B. The conversion in Model A refers to the long alkyl chains (R-C_nH_{2n+1}, R- is the kerogen core); in Model B, all species are included because they all are precursors of CH₄.

During the revision, we used the same definition of conversion in both models: the ratio between the ¹²C atoms in CH₄ to the ¹²C atoms in all the alkyl and alkane species. Consequently, the conversion accounting for the Kidd Creek gas composition is about 0.65-0.70 (Fig. 3A). We presented the Kidd Creek gas data with this range (0.65-0.70), instead of 0.95 – 1.0 in previous versions. After a slight modification of kinetic parameters (k_b/k_a changes from 10 to 20 to match the measured δD value; α_{ka001} and α_{kb001} changed from 1.15 to 1.10 to match the measured $\Delta^{12}CH_2D_2$ value), the modeled isotopic compositions are still located within the range of Kidd Creek gas data.

Reviewer 2 might have an impression that there is only a very narrow conversion window that can make the simulation results close enough to the lab data, an impression from the early version of this work when we did not use inverse DKIE and ^{13}C KIE. Results from Models A and B repeatedly indicate the presence of inverse KIE. Once inverse KIEs are applied, Model B is highly underdetermined, there is a wide conversion window to satisfy the model fitting.

Comment: *The second disagreement concerns the carbon isotope composition of methane. Model A predicts that as the conversion fraction increases from 0 to 0.85, the cumulative methane $d^{13}\text{C}$ value lowers by just 1.5 ‰. However, model B predicts that over the same amount of reaction progress, the cumulative methane $d^{13}\text{C}$ value lowers by twice that amount, 3 ‰.*

Reply: We thank the reviewer for pointing out the inconsistency – this adds a new constraint on the underdetermined Model B. By changing ^{13}C KIE parameters α_{ka100} and α_{kb100} from 1.0015 to 1.0009, the decrease of methane $\delta^{13}\text{C}$ is about 1.2‰ in both models when conversion increases from 0 to 0.7. The α_{ka100} and α_{kb100} here are different from the values obtained from Model A (1.0015) because, as the reviewer acknowledged, Model B does not include a normal secondary ^{13}C KIE. We added a note on this point after Table 1.

Comment: *At high degrees of reaction progress (>0.8, such as model B requires), nearly all of the intermediate R-CH₃ pool should reside in ethane. Therefore, the $d^{13}\text{C}$ value of the R-CH₃ at a conversion fraction of 0.8 in model B should closely approximate the $d^{13}\text{C}$ value of ethane in model A at the same conversion fraction. These results are not reported, so this comparison cannot be evaluated. However, it is likely that these predictions strongly disagree. This is because model B does not simulate secondary carbon isotope effects of C-C cleavage, but model A does.*

Reply: Following this suggestion we calculated the $\delta^{13}\text{C}$ of R-CH₃ and obtained -35.5‰ at the conversion fraction of 0.7, not far from $\delta^{13}\text{C} = -36.6\%$ of C₂H₆ at this conversion. Note that at this conversion there is abundant species other than C₂H₆ as the precursors of CH₄, so the small difference does not indicate inconsistency (although Model B cannot simulate secondary ^{13}C KIE). We submitted the time-lapse abundance of all the species from both models for review.

Reply to Reviewer 4

We thank Reviewer 4's detailed comments, insightful evaluation, and constructive suggestions on the work. Reviewer 4 suggested us include four points in the work: 1) using "Chung"-like plots; 2) simulating CH₂D₂ fractionation, specifically, kinetic relaxation (isotope exchange between methane isotopologues to obtain an equilibrated $\Delta^{12}\text{CH}_2\text{D}_2$); 3) using O₂ fugacity or PAL to describe the redox condition; and 4) evaluating diffusive isotopic fractionation over 3 Gyr.

We already applied the Chung-type plots to discuss the observation in the previous version. Following the above suggestions, we included Archean O₂ PAL number, discussion on diffusive fractionation, and discussion on kinetic relaxation in the revised version. Below are the point-to-point replies to the reviewer's comments.

Comment: *Summary/Key Results:*

In this study the authors analyze previously generated carbon and hydrogen isotope data from the Kidd Creek Mine, from a Neoproterozoic Greenstone Belt in Ontario, Canada. The significance of the samples, is that they are some of the oldest occurring hydrocarbons and they have very depleted hydrogen isotope values, making their interpretation difficult. Previous studies have suggested that the complex isotope patterns are a result of Fischer-Tropsch synthesis, but these authors suggest it is a biotic signature. The authors call on hydrogenotrophy using the alkyl chains to help in the processes as the major metabolism imparting the very depleted hydrogen isotope values in this system. If true, this would be of great significance because it may be the oldest record of hydrogenotrophic metabolism preserved in a geological record.

Reply: We appreciate the reviewer's insightful summary. These are the exact points we would like to present in the work.

Comment: *Validity: There are some large logical leaps which need further justification. While there is no strong evidence that counters the conclusions that the authors draw, there are more simple explanations that the authors have not considered and countered.*

Reply: While any simple explanations may separately explain one or two isotopic signatures, they often contradict each other when putting all the geochemical signatures together. A

consistent interpretation on all the published data is the most challenging task that this work achieved, as partly can be seen in our reply to Reviewer 2's comments.

Comment: *The major concern with the study is that the biologic origin of the $d^{13}C$ pattern is not sufficiently justified. The authors claim that resultant isotope fingerprint can only occur as a result of metabolism. However, the most canonical treatment of hydrocarbons (Chung 1988 and many works that follow) allows for a much simpler explanation of the data. In general, it is predicted for a linear decrease in $d^{13}C$ vs $1/n$ (where n is the carbon number of the alkane in the system). If the methane deviates from the line of butane, propane, ethane below it is thought to be due to microbial activity. If it deviates above (like it seems as if this data does) it is thought to be from diffusive loss.*

Reply: We presented the Chung-type plot for carbon isotopes as Fig. 5D; Chung-type plot for hydrogen isotopes as Figs. 1C and 5C. The Chung-type plot of Fig. 5D shows a narrow isotopic fractionation between alkanes in the Kidd Creek gas; the slope is completely different from thermogenic gas and cannot be explained with the mixing of microbial methane. If it were caused by diffusive loss, then the slope in Fig. 1C (hydrogen isotopes) could not be explained. We evaluated the impact of diffusive loss on isotopic compositions in the revised version (page 9).

Comment: *While this deviation is much larger than in other studies, there could be diffusive loss of methane for ~3 billion years. While this would indicate a very large initial reservoir of methane, it should be calculated and modeled. This is a very traditional and useful representation of hydrocarbon isotope values, and as it would also be a very simple and intuitive answer to the complex data, it needs to be ruled out before more complex interpretation can continue.*

Reply: We evaluated and discussed the diffusive loss in the revised version (page 9); the diffusive fractionation does not have significant impact on the isotopic compositions of the Kidd Creek gas (because of the low diffusivity of the crystalline rocks).

Comment: *There is another line of evidence that the others propose which can be explained more simply. The authors use the $\Delta^{12}CH_2D_2$ near zero as evidence of biological fractionation (when most biological fractionation significantly depletes this measurement. Due to the*

extremely long storage time, many reactions that are kinetically limited can occur (like the re-ordering of deuterium isotopes between methane molecules). There is a recent study by Labidi et al looking at the relative $^{13}\text{CH}_3\text{D}$ vs $^{12}\text{CH}_2\text{D}_2$ of methane at mid ocean ridges, that show that the $^{12}\text{CH}_2\text{D}_2$ can lose its depletion relative to expected equilibrium very quickly. The authors use their simple kinetic model of a $\Delta^{12}\text{CH}_2\text{D}_2 \sim 0$ to indicate their preferred methane metabolism, when other authors have shown significant relaxation of the $\Delta^{12}\text{CH}_2\text{D}_2$ depletion on “modern” timescales at vents. While this not be what is occurring in Kidd Creek, it needs to be addressed as it is the simpler explanation to the isotope value.

Reply: We noticed some recent papers using equilibrating process to explain clumped isotopes of methane. We modeled this exchange in a previous work (Xia and Gao, GCA 2019) and found that the equilibrating process requires hydrogen atom exchange between methane molecules; it further requires high thermal stress (the hydrocarbon gas should be in dry gas stage in thermal maturity) and catalysts (free transitional metal in the lab and free radicals under geological conditions). As we have addressed, *“in other words, ethane and propane will decompose when methane isotope exchange occurs. This can happen mainly at high thermal maturity”* (Xia and Gao, GCA 2019, section 5.2). The high fraction of C_{2+} alkanes (15 vol%) in the Kidd Creek gas rules out a significant hydrogen exchange between methane isotopologues in the Kidd Creek gas.

For revision, we included the discussion on why the kinetic relaxation is not a suitable explanation.

Comment: *The other easy justification of the $\Delta^{12}\text{CH}_2\text{D}_2$ value is the combinatorial effect, which has been documented theoretically and experimentally in recent studies. This allows for a range of clumped values to occur with depleted hydrogen, especially if there are two disparate pools of hydrogen contributing to the methane isotope value.*

Reply: While a depletion of $^{12}\text{CH}_2\text{D}_2$ reflects combinatorial effect (and therefore indicates two H sources), a reaction with two H sources do not always show a combinatorial effect. This has been discussed in our previous paper (Xia and Gao, GCA 2019, Fig. 6F).

Comment: *As the dD of the methane is not as low as the dD of the H_2 from serpentinization, this would indicate that the hydrogen addition could in addition to the inheritance from the precursor molecule, could also come from two disparate pools.*

Reply: We appreciate the reviewer for concurring with the key point of this work.

Comment: *Furthermore, while oxygen was likely quite low in the Neoproterozoic, the authors need to indicate the oxygen range that they are speaking about in order for this to add to the discussion of early earth surface conditions. The authors say that it had to be “low” in order for the metabolism they call on to occur, and that this does not happen in more recent serpentinization environments. As this argument hinges on the low pO_2 of the Archean it is necessary to explain the oxygen threshold (either in PAL or fugacity as it is within the crust) that these reactions occur at. Also, serpentinites have a range of fugacity's (and resultant Fe^{3+}/Fe^{2+} values) as is dependent on the fluids flushing the system. Therefore, it is more important to declare a value as the range is large.*

Reply: In the revised version, we cited the O_2 partial pressure and sulfate concentration relative to present levels to describe the redox condition (Page 8).

Comment: *Finally, strongly reconsider the title away from GoE and focus on interpretation of the deep mine methane in context to the geochemical environment of the mine system and diagenetic processes (see Giunta et al.; Sherwood-Lollar et al. and other studies of bulk and clumped isotope systematics from those samples).*

Reply: At the beginning of the preparation this manuscript in 2018 we intended to focus on the point that the Kidd Creek gas is not abiotic. After extensive discussions with reviewers and editors of *Natural Geoscience*, we realized that the pre-GOE ecosystem and elemental cycle as reflected by our interpretation have more scientific significance, that is how the title was finalized. As expressed in the manuscript and in the replies to the reviewer's comments, the evidence is solid to support the arguments. We have enhanced the discussion by addressing the reviewer's suggestions.

Although some previous works (Giunta et al., 2019; Sherwood Lollar et al., 2007) discussed the microbial activities influencing the Kidd Creek gas isotopes, they did not clearly address that Kidd Creek gas is not abiotic.

Comment: *Significance: If the conclusions of the paper are true, then the paper does have sufficient significance to be published in *Nature Communications*. Evidence of metabolism that predate the GOE are always very important to understanding the rise of life on Earth. In*

addition, this metabolism can only occur in very low oxygen environments which also further adds to earth surface constraints at a time when little data exists. Finally, the presence of actual biologically produced products that are Neoproterozoic in age is also significant by itself.

Reply: Again, we appreciate the reviewer's assessment on the significance of this work.

Comment: *Data and Methodology: The study does not create any data. All measurements listed in the figures are from previously published work. This is somewhat disappointing, as there are a lot of new measurements in methane isotopes that could perhaps further constrain this complex system. While the authors do bring in the clumped isotope results that already exist from this well studied local (including both $^{13}\text{CH}_3\text{D}$ and $^{12}\text{CH}_2\text{D}_2$), their indicators could be taken more into account.*

Reply: For decades, the Kidd Creek gas has accumulated the most abundant data of various measurements (gas composition, carbon and hydrogen isotopic compositions, and clumped isotopic compositions of methane) among all alkane gas accumulations except for commercial natural gas reservoirs. A consistent interpretation on all the published geochemical and geological data is more demanding than adding new data points; and such an interpretation is one of the key significances of this work.

Comment: *Analytical Results: The authors do generate a novel model with available published data, and provide a new interpretive framework for samples from systems such as the Kidd Creek mine. The model approach could be extrapolated to other available environmental data (see recent studies from Labidi et al.; Giunta et al.; and others). Specifically, diffusive modeling and kinetic relaxation of the depleted $\Delta^{12}\text{CH}_2\text{D}_2$ would be beneficial additions. In addition, the Chung-type plots as requested above would help the reader place into context the new advance made herein.*

Reply: we thank again for the reviewer's assessment on this work. For the issues of Chung-type plot, relaxation of $^{12}\text{CH}_2\text{D}_2$ (through hydrogen exchange between methane isotopologues), we replied on pages 4 and 5 of this "Reply to Reviewers Comments" Report.

Comment: *Suggested improvements:*

The paper needs three major additions: a) “Chung”-like diagrams made (or fig 1 modified) such that traditional interpretation of the carbon isotope values can be completed easily (if it does not have conclusive results can be included in supplement).

Reply: The previous version already included Chung-type plots (Figs. 1C and 5C; Fig. 5D). Also see the reply on page 4 of this document.

Comment: *b) Modeling of diffusive loss of methane and its effect on the deuterium isotope value.*

Reply: We included the assessment on diffusive fractionation in the revised version (Page 9).

Comment: *c) Modeling of kinetic relaxation of $\Delta^{12}\text{CH}_2\text{D}_2$ clumping in methane*

Reply: The reviewer meant modeling the isotope exchange between methane isotopologues. This issue was replied on page 5 of this document.

Comment: *d) Discussion of oxygen fugacity or at least relative to PAL.*

Reply: In the revised version we included O_2 pressure relative to PAL to describe the redox conditions.

Comment: *Clarity and context: The text needs to be proofread. There are many errors that decrease clarity of the arguments. Specific examples are included in the line comments, but in general the language could use work.*

Reply: A native English speaker has proofread and copy edited the revised version.

Comment: *References: Sufficient, though there are some more recent papers emerging from both the Eiler lab at CalTech and the Young lab at UCLA that have dealt with methane from a variety of environments.*

Reply: We read and reviewed several recent clumped isotope papers and added necessary literature review in this work.

Comment: *Expertise:*

A. Piasceki: Stable isotope geochemist that has done work on site-specific and clumped isotopes within small n-alkanes from modeling, methods development, and measurement.

B. W. Leavitt: Geomicrobiology & microbial geochemistry.

Reply: We noticed the recent works of A. Piasecki and W. Leavitt on specific isotopes although did not find suitable places in the manuscript to cite their works.

Comment: *Specific Line comments*

Line 71: why do you have 2 hydrogens per cleavage and not one (theoretically this is the same a thermogenic gas generation over long term, how specifically is this different? unclear besides the incredibly depleted H₂ value from serpentinization specifically why biology must be introduced) (discussed above as well)

Reply: The mass balance is $R-CH_2-CH_2-R' + 2 [H] \rightarrow R-CH_3 + CH_3-R'$, therefore two [H] atoms per cleavage are needed. Biological factor is discussed in the following section “Microbial hydrogenolysis suggested by ¹³C and ¹²CH₂D₂ distributions” in the manuscript.

Comment: *Line 92 : sentence structure is ambiguous “because organic precursor is the single carbon source” what does this mean, do you have a single organic carbon precursor (if yes how do you know this, there is usually a diversity of molecules even if from one environment), and the sentence continues to say “kinetic isotope fractionation is the main reason for ... variation within short alkanes”, which isn’t strictly true, but it does emphasize preexisting structures within the precursor molecules, and overprints an isotope fractionation on top of that*

Reply: Here we wanted to express that while H atoms in the alkanes are from two sources (H₂ and alkyl groups), the C atoms are only from the alkyl group. For revision, the sentence is changed to “In contrast to the hydrogen atoms, which have two sources, the carbon atoms in the Kidd Creek gas hydrocarbon gases are all from organic precursors. Therefore, kinetic isotopic fractionation, instead of mixing, is the main reason for the δ¹³C variation between short-chain alkanes”.

Comment: *line 101: does not fully justify the biology of cracking, following the early stolper papers we have some generation of methane at high temperatures, its just how long are the samples within the cracking window, yes you still have higher n-alkanes but this doesn't rule out thermogenic generation?*

Reply: If the Kidd Creek gas were in the cracking window for a long time, we would have seen a typical thermogenic gas signature on the Chung-like plots. The deviation from the Chung-type plots cannot be explained with diffusive fractionation (added in the revised version). After

putting all the puzzles together, a microbial hydrogenolysis is the only consistent explanation we can find.

Comment: *line 131 doesn't make sense "where oxidizers that are more efficient than alkanes" are alkanes oxidizers? or is it what the bugs do?*

Reply: "oxidizers" changed to "oxidizing chemical species".

Comment: *line 156: sentence needs modification (became more positive years after boreholes were drilled? need verb)*

Reply: "drilled" added.

Comment: *following sentence "microbes has" -> microbes have*

Reply: revised.

Comment: *Reference Cited: Chung, H.M., Gormly, J.R. and Squires, R.M., 1988. Origin of gaseous hydrocarbons in subsurface environments: theoretical considerations of carbon isotope distribution. Chemical Geology, 71(1-3), pp.97-104.*

Reply: This significant paper was cited from the beginning of the preparation of this manuscript (reference 20 of the previous submission).

REVIEWERS' COMMENTS

Reviewer #2 (Remarks to the Author):

This my fifth review of this manuscript. In my last review, I remarked that the authors were presenting two models that made incompatible predictions for the isotope compositions of the Kidd Creek hydrocarbons. In the latest revision, the authors have changed the parameters of model B so that the two models are now compatible. This was an important advance and substantially improves the quality of the work. I have no further concerns with the models or interpretation. In my view, this manuscript should be accepted. It will be of broad significance and is likely to spur future work on microbial hydrogenolysis.